# Learn to Pay Attention

**Saumya Jetley, Nicholas A. Lord, Namhoon Lee & Philip H. S. Torr**
Department of Engineering Science, University of Oxford
{sjetley,nicklord,namhoon,phst}@robots.ox.ac.uk

## Abstract

We propose an end-to-end-trainable attention module for convolutional neural network (CNN) architectures built for image classification. The module takes as input the 2D feature vector maps which form the intermediate representations of the input image at different stages in the CNN pipeline, and outputs a 2D matrix of scores for each map. Standard CNN architectures are modified through the incorporation of this module, and trained under the constraint that a convex combination of the intermediate 2D feature vectors, as parameterised by the score matrices, must *alone* be used for classification. Incentivised to amplify the relevant and suppress the irrelevant or misleading, the scores thus assume the role of attention values. Our experimental observations provide clear evidence to this effect: the learned attention maps neatly highlight the regions of interest while suppressing background clutter. Consequently, the proposed function is able to bootstrap standard CNN architectures for the task of image classification, demonstrating superior generalisation over 6 unseen benchmark datasets. When binarised, our attention maps outperform other CNN-based attention maps, traditional saliency maps, and top object proposals for weakly supervised segmentation as demonstrated on the Object Discovery dataset. We also demonstrate improved robustness against the fast gradient sign method of adversarial attack.

## 1 Introduction

Feed-forward convolutional neural networks (CNNs) have demonstrated impressive results on a wide variety of visual tasks, such as image classification, captioning, segmentation, and object detection. However, the visual reasoning which they implement in solving these problems remains largely inscrutable, impeding understanding of their successes and failures alike.

One approach to visualising and interpreting the inner workings of CNNs is the attention map: a scalar matrix representing the relative importance of layer activations at different 2D spatial locations with respect to the target task (Simonyan et al., 2013). This notion of a nonuniform spatial distribution of relevant features being used to form a task-specific representation, and the explicit scalar representation of their relative relevance, is what we term 'attention'. Previous works have shown that for a classification CNN trained using image-level annotations alone, extracting the attention map provides a straightforward way of determining the location of the object of interest (Cao et al., 2015; Zhou et al., 2016) and/or its segmentation mask (Simonyan et al., 2013), as well as helping to identify discriminative visual properties across classes (Zhou et al., 2016). More recently, it has also been shown that training smaller networks to mimic the attention maps of larger and higher-performing network architectures can lead to gains in classification accuracy of those smaller networks (Zagoruyko & Komodakis, 2016).

The works of Simonyan et al. (2013); Cao et al. (2015); Zhou et al. (2016) represent one series of increasingly sophisticated techniques for estimating attention maps in classification CNNs. However, these approaches share a crucial limitation: all are implemented as post-hoc additions to fully trained networks. On the other hand, integrated attention mechanisms whose parameters are learned over the course of end-to-end training of the entire network have been proposed, and have shown benefits in various applications that can leverage attention as a cue. These include attribute prediction (Seo et al., 2016), machine translation (Bahdanau et al., 2014), image captioning (Xu et al., 2015; You et al., 2016; Mun et al., 2016) and visual question answering (VQA) (Xu & Saenko, 2016; Yang et al., 2016). Similarly to these approaches, we here represent attention as a probabilistic map over

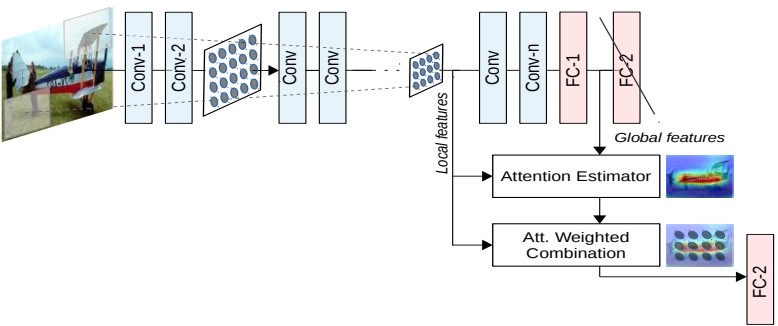

Figure 1: Overview of the proposed attention mechanism.

the input image locations, and implement its estimation via an end-to-end framework. The novelty of our contribution lies in repurposing the global image representation as a query to estimate multi-scale attention in classification, a task which, unlike *e.g.* image captioning or VQA, does not naturally involve a query.

Fig. 1 provides an overview of the proposed method. Henceforth, we will use the terms 'local features' and 'global features' to refer to features extracted by some layer of the CNN whose effective receptive fields are, respectively, contiguous proper subsets of the image ('local') and the entire image ('global'). By defining a compatibility measure between local and global features, we redesign standard architectures such that they must classify the input image using *only* a weighted combination of local features, with the weights represented here by the attention map. The network is thus forced to learn a pattern of attention relevant to solving the task at hand.

We experiment with applying the proposed attention mechanism to the popular CNN architectures of VGGNet (Simonyan & Zisserman, 2014) and ResNet (He et al., 2015), and capturing coarse-to-fine attention maps at multiple levels. We observe that the proposed mechanism can bootstrap baseline CNN architectures for the task of image classification: for example, adding attention to the VGG model offers an accuracy gain of $7\%$ on CIFAR-100. Our use of attention-weighted representations leads to improved fine-grained recognition and superior generalisation on 6 benchmark datasets for domain-shifted classification. As observed on models trained for fine-grained bird recognition, attention aware models offer limited resistance to adversarial fooling at low and moderate $L_\infty$-noise norms. The trained attention maps outperform other CNN-derived attention maps (Zhou et al., 2016), traditional saliency maps ( Jiang et al. (2013); Zhang & Sclaroff (2013)), and top object proposals on the task of weakly supervised segmentation of the Object Discovery dataset ( Rubinstein et al. (2013)). In §5, we present sample results which suggest that these improvements may owe to the method's tendency to highlight the object of interest while suppressing background clutter.

## 2 RELATED WORK

*Attention in CNNs* is implemented using one of two main schemes - post hoc network analysis or trainable attention mechanisms. The former scheme has been predominantly employed to access network reasoning for the task of visual object recognition (Simonyan et al., 2013; Zhou et al., 2016; Cao et al., 2015). Simonyan et al. (2013) approximate CNNs as linear functions, interpreting the gradient of a class output score with respect to the input image as that class's spatial support in the image domain, *i.e.* its attention map. Importantly, they are one of the first to successfully demonstrate the use of attention for localising objects of interest using image-level category labels alone. Zhou et al. (2016) apply the classifier weights learned for image-level descriptors to patch descriptors, and the resulting class scores are used as a proxy for attention. Their improved localisation performance comes at the cost of classification accuracy. Cao et al. (2015) introduce attention in the form of binary nodes between the network layers of Simonyan et al. (2013). At test time, these attention maps are adapted to a fully trained network in an additional parameter tuning step. Notably, all of the above methods extract attention from fully trained CNN classification models, *i.e.* via post-processing. Subsequently, as discussed shortly, many methods have explored the per-

formance advantages of optimising the weights of the attention unit in tandem with the original network weights.

*Trainable attention in CNNs* falls under two main categories - hard (stochastic) and soft (deterministic). In the former, a hard decision is made regarding the use of an image region, often represented by a low-order parameterisation, for inference (Mnih et al., 2014; Xu et al., 2015). The implementation is non-differentiable and relies on a sampling-based technique called REINFORCE for training, which makes optimising these models more difficult. On the other hand, the soft-attention method is probabilistic and thus amenable to training by backpropagation. The method of Jaderberg et al. (2015) lies at the intersection of the above two categories. It uses a parameterised transform to estimate hard attention on the input image deterministically, where the parameters of the image transformation are estimated using differentiable functions. The soft-attention method of Seo et al. (2016) demonstrates improvements over the above by implementing nonuniform non-rigid attention maps which are better suited to natural object shapes seen in real images. It is this direction that we explore in our current work.

*Trainable soft attention in CNNs* has mainly been deployed for query-based tasks (Bahdanau et al., 2014; Xu et al., 2015; Seo et al., 2016; You et al., 2016; Mun et al., 2016; Xu & Saenko, 2016; Yang et al., 2016). As is done with the exemplar captions of Mun et al. (2016), the questions of Xu & Saenko (2016); Yang et al. (2016), and the source sentences of Bahdanau et al. (2014), we here map an image to a high-dimensional representation which in turn highlights the relevant parts of the input image to guide the desired inference. We draw a close comparison to the progressive attention approach of Seo et al. (2016) for attribute prediction. However, there are some noteworthy differences. Their method uses a one-hot encoding of category labels to query the image: this is unavailable to us and we hence substitute a learned representation of the global image. In addition, their sequential mechanism refines a single attention map along the length of the network pipeline. This doesn't allow for the expression of a complementary focus on different parts of the image at different scales as leveraged by our method, illustrated for the task of fine-grained recognition in §5.

*The applications of attention,* in addition to facilitating the training task, are varied. The current work covers the following areas:

· *Domain shift*: A traditional approach to handling domain shift in CNNs involves fine-tuning on the new dataset, which may require thousands of images from each target category for successful adaptation. We position ourselves amongst approaches that use attention (Zhou et al., 2016; Jaderberg et al., 2015) to better handle domain changes, particularly those involving background content, occlusion, and object pose variation, by selectively focusing to the objects of interest.

· *Weakly supervised semantic segmentation*: This area investigates image segmentation using minimal annotations in the form of scribbles, bounding boxes, or image-level category labels. Our work uses category labels and is related to the soft-attention approach of Hong et al. (2016). However, unlike the aforementioned, we do not explicitly train our model for the task of segmentation using any kind of pixel-level annotations. We evaluate the binarised spatial attention maps, learned as a by-product of training for image classification, for their ability to segment objects.

· *Adversarial robustness*: The work by Goodfellow et al. (2014) explores the ease of fooling deep classification networks by adding an imperceptible perturbation to the input image, implemented as an epsilon step in the direction opposite to the predicted class score gradient. The works by Wang et al. (2016) and Gao et al. (2017) argue that this vulnerability comes from relying on spurious or non-oracle features for classification. Consequently, Gao et al. (2017) demonstrate increased adversarial robustness by identifying and masking the likely adversarial feature dimensions. We experiment with performing such suppression in the spatial domain.

## 3 APPROACH

The core goal of this work is to use attention maps to identify and exploit the effective spatial support of the visual information used by CNNs in making their classification decisions. This approach is premised on the hypothesis that there is benefit to identifying salient image regions and amplifying their influence, while likewise suppressing the irrelevant and potentially confusing information in other regions. In particular, we expect that enforcing a more focused and parsimonious use of image information should aid in generalisation over changes in the data distribution, as occurs for instance when training on one set and testing on another. Thus, we propose a trainable attention estimator and illustrate how to integrate it into standard CNN pipelines so as to influence their output as outlined

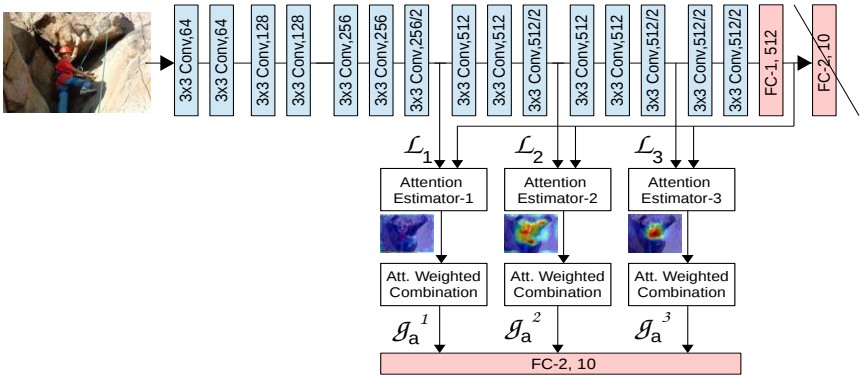

Figure 2: Attention introduced at 3 distinct layers of VGG. Lowest level attention maps appear to focus on the surroundings (*i.e.*, the rocky mountain), intermediate level maps on object parts (*i.e.*, harness and climbing equipment) and the highest level maps on the central object.

above. The method is based on enforcing a notion of compatibility between local feature vectors extracted at intermediate stages in the CNN pipeline and the global feature vector normally fed to the linear classification layers at the end of the pipeline. We implement attention-aware classification by restricting the classifier to use *only* a collection of local feature vectors, as chosen and weighted by the compatibility scores, in classifying examples. We will first discuss the modification to the network architecture and the method of training it, given a choice of compatibility function. We will then conclude the method description by presenting alternate choices of the compatibility function.

## 3.1 DESIGN AND TRAINING OF ATTENTION SUBMODULE

The proposed approach is illustrated in Fig. 2. Denote by $\mathcal{L}^s = \{\boldsymbol{\ell}_1^s, \boldsymbol{\ell}_2^s, \cdots, \boldsymbol{\ell}_n^s\}$ the set of feature vectors extracted at a given convolutional layer $s \in \{1, \cdots, S\}$. Here, each $\boldsymbol{\ell}_i^s$ is the vector of output activations at the spatial location $i$ of $n$ total spatial locations in the layer. The global feature vector $\boldsymbol{g}$ has the entire input image as support and is output by the network's series of convolutional and nonlinear layers, having only to pass through the final fully connected layers to produce the original architecture's class score for that input. Assume for now the existence of a compatibility function $\mathcal{C}$ which takes two vectors of equal dimension as arguments and outputs a scalar compatibility score: this will be specified in 3.2.

The method proceeds by computing, for each of one or more layers $s$, the set of compatibility scores $\mathcal{C}(\hat{\mathcal{L}}^s, \boldsymbol{g}) = \{c_1^s, c_2^s, \dots c_n^s\}$, where $\hat{\mathcal{L}}^s$ is the image of $\mathcal{L}^s$ under a linear mapping of the $\boldsymbol{\ell}_i^s$ to the dimensionality of $\boldsymbol{g}$. The compatibility scores are then normalised by a softmax operation:

$$a_i^s = \frac{\exp(c_i^s)}{\sum_j^n \exp(c_j^s)}, \ i \in \{1 \cdots n\}. \tag{1}$$

The normalised compatibility scores $\mathcal{A}^s = \{a_1^s, a_2^s, \cdots a_n^s\}$ are then used to produce a single vector $\boldsymbol{g}_a^s = \sum_{i=1}^n a_i^s \cdot \boldsymbol{\ell}_i^s$ for each layer $s$, by simple element-wise weighted averaging. Crucially, the $\boldsymbol{g}_a^s$ now *replace* $\boldsymbol{g}$ as the global descriptor for the image. For a network trained under the restriction that the $\boldsymbol{g}_a^s$ alone are used to classify the input image, $\mathcal{A}$ corresponds to 'attention' as defined earlier.

In the case of a single layer ($S = 1$), the attention-incorporating global vector $\boldsymbol{g}_a$ is computed as described above, then mapped onto a $T$-dimensional vector which is passed through a softmax layer to obtain class prediction probabilities $\{\hat{p}_1, \hat{p}_2, \cdots \hat{p}_T\}$, where $T$ is the number of target classes. In the case of multiple layers ($S > 1$), we compare two options: concatenating the global vectors into a single vector $\boldsymbol{g}_a = [\boldsymbol{g}_a^1, \boldsymbol{g}_a^2, \cdots \boldsymbol{g}_a^S]$ and using this as the input to the linear classification step as above, or, using $S$ different linear classifiers and averaging the output class probabilities. All free network parameters are learned in end-to-end training under a cross-entropy loss function.

## 3.2 Choice of compatibility function $\mathcal{C}$

The compatibility score function $\mathcal{C}$ can be defined in various ways. The alignment model from Bahdanau et al. (2014); Xu et al. (2015) can be re-purposed as a compatibility function as follows:

$$c_i^s = \langle \boldsymbol{u}, \boldsymbol{\ell}_i^s + \boldsymbol{g} \rangle, \; i \in \{1 \cdots n\}, \tag{2}$$

Given the existing free parameters between the local and the global image descriptors in a CNN pipeline, we can simplify the concatenation of the two descriptors to an addition operation, without loss of generality. This allows us to limit the parameters of the attention unit. We then learn a single fully connected mapping from the resultant descriptor to the compatibility scores. Here, the weight vector $\boldsymbol{u}$ can be interpreted as learning the universal set of features relevant to the object categories in the dataset. In that sense, the weights may be seen as learning the general concept of objectness.

Alternatively, we can use the dot product between $\boldsymbol{g}$ and $\boldsymbol{\ell}_i^s$ as a measure of their compatibility:

$$c_i^s = \langle \boldsymbol{\ell}_i^s, \boldsymbol{g} \rangle, \; i \in \{1 \cdots n\}. \tag{3}$$

In this case, the relative magnitude of the scores would depend on the alignment between $\boldsymbol{g}$ and $\boldsymbol{\ell}_i^s$ in the high-dimensional feature space and the strength of activation of $\boldsymbol{\ell}_i^s$.

## 3.3 Intuition

In a standard CNN architecture, a global image descriptor $\boldsymbol{g}$ is derived from the input image and passed through a fully connected layer to obtain class prediction probabilities. The network must express $\boldsymbol{g}$ via mapping of the input into a high-dimensional space in which salient higher-order visual concepts are represented by different dimensions, so as to render the classes linearly separable from one another. Our method encourages the filters *earlier* in the CNN pipeline to learn similar mappings, compatible with the one that produces $\boldsymbol{g}$ in the original architecture. This is achieved by allowing a local descriptor $\boldsymbol{\ell}_i$ of an image patch to contribute to the final classification step only in proportion to its compatibility with $\boldsymbol{g}$ as detailed above. That is, $\mathcal{C}(\hat{\boldsymbol{\ell}}_i, \boldsymbol{g})$ should be high if and only if the corresponding patch contains parts of the dominant image category. Note that this implies that the effective filters operating over image patches in the layers $s$ must represent relatively 'mature' features with respect to the classification goal. We thus expect to see the greatest benefit in deploying attention relatively late in the pipeline. Further, different kinds of class details are more easily accessible at different scales. Thus, in order to facilitate the learning of diverse and complementary attention-weighted features, we propose the use of attention over different spatial resolutions. The combination of the two factors stated above results in our deploying the attention units after the convolutional blocks that are late in the pipeline, but before their corresponding max-pooling operations, *i.e.* before a drop in the spatial resolution. Note also that the use of the softmax function in normalising the compatibility scores enforces $0 \le a_i \le 1 \; \forall i \in \{1 \cdots n\}$ and $\sum_i a_i = 1$, *i.e.* that the combination of feature vectors is convex. This ensures that features at different spatial locations must effectively compete against one another for their share of the attention map. The compatibility scores thus serve as a robust proxy for attention in the classification pipeline.

## 4 Experimental Setup

To incorporate attention into the VGG network, we move each of the first 2 max-pooling layers of the baseline architecture after each of the 2 corresponding additional convolutional layers that we introduce at the end of the pipeline. By pushing the pooling operations further down the pipeline, we ensure that the local layers used for estimating attention have a higher resolution. Our modified model has 17 layers: 15 convolutional and 2 fully connected. The output activations of layer-16 (fc) define our global feature vector $\boldsymbol{g}$. We use the local feature maps from layers 7, 10, and 13 (convolutional) for estimating attention. We compare our approach with the activation-based attention method of Zhou et al. (2016), and the progressive attention mechanism of Seo et al. (2016). For RNs (He et al., 2015), we use a 164-layered network. We replace the spatial average-pooling step after the computational block-4 with extra convolutional and max-pooling steps to obtain the global feature $\boldsymbol{g}$. The outputs of blocks 2, 3, and 4 serve as the local descriptors for attention. For more details about network architectures refer to §A.2. Note that, for both of the above architectures, if the dimensionality of $\boldsymbol{g}$ and the local features of a layer $s$ differ, we project $\boldsymbol{g}$ to the lower-dimensional

space of the local features, instead of the reverse. This is done in order to limit the parameters at the classification stage. The global vector $g$, once mapped to a given dimensionality, is then shared by the local features from different layers $s$ as long as they are of that dimensionality.

We refer to the network *Net* with attention at the last level as *Net-att*, at the last two levels as *Net-att2*, and at the last three levels as *Net-att3*. We denote by *dp* the use of the dot product for matching the global and local descriptors and by *pc* the use of parametrised compatibility. We denote by *concat* the concatenation of descriptors from different levels for the final linear classification step. We use *indep* to denote the alternative of independent prediction of probabilities at different levels using separate linear classifiers: these probabilities are averaged to obtain a single score per class.

We evaluate the benefits of incorporating attention into CNNs for the primary tasks of image classification and fine-grained object category recognition. We also examine robustness to the kind of adversarial attack discussed by Goodfellow et al. (2014). Finally, we study the quality of attention maps as segmentations of image objects belonging to the network-predicted categories. For details of the datasets used and their pre-processing routines refer to §A.1 , for network training schedules refer to §A.3 , and for task-specific experimental setups refer to §A.4.

## 5 RESULTS AND DISCUSSION

### 5.1 IMAGE CLASSIFICATION AND FINE-GRAINED RECOGNITION

| Model | Top-1 error with standard deviation. | |
| --- | --- | --- |
| | CIFAR-10 | CIFAR-100 |
| — Existing architectures — | | |
| VGG (Simonyan & Zisserman, 2014) | 7.77 (0.08) | 30.62 (0.16) |
| VGG-GAP (Zhou et al., 2016) | 9.87 (0.10) | 31.77 (0.13) |
| VGG-PAN (Seo et al., 2016) | 6.29 (0.03) | **24.35 (0.14)** |
| RN-164 (He et al., 2015) | **6.03 (0.18)** | 25.34 (0.16) |
| — Architectures with attention — | | |
| (VGG-att)-dp | 6.14 (0.06) | 24.22 (0.08) |
| (VGG-att2)-indep-dp | 5.91 (0.05) | 23.24 (0.07) |
| (VGG-att2)-concat-dp | 5.86 (0.05) | 23.91 (0.11) |
| (VGG-att)-pc | 5.67 (0.04) | 23.70 (0.07) |
| (VGG-att2)-indep-pc | 5.36 (0.06) | 24.00 (0.06) |
| (VGG-att2)-concat-pc | **5.23 (0.04)** | 23.19 (0.04) |
| (VGG-att3)-concat-pc | 6.34 (0.07) | **22.97 (0.04)** |

Table 1: CIFARs: Top-1 classification errors.

| Model | Top-1 error with standard deviation. | |
| --- | --- | --- |
| | CUB-200-2011 | SVHN |
| — Existing architectures — | | |
| VGG (Simonyan & Zisserman, 2014) | 34.64 (0.26) | **4.27 (0.04)** |
| VGG-GAP (Zhou et al., 2016) | 29.50* ( – ) | 5.84 (0.09) |
| VGG-PAN (Seo et al., 2016) | 31.46 (0.16) | 8.02 (0.06) |
| RN-34 (Zagoruyko & Komodakis, 2016) | **26.5* (–)** | – |
| — Architectures with attention — | | |
| (VGG-att2)-concat-pc | **26.80 (0.16)** | 3.74 (0.05) |
| (VGG-att3)-concat-pc | 26.95 (0.10) | **3.52 (0.04)** |

Table 2: Fine-grained recognition: Top-1 errors. * denotes results from publications.

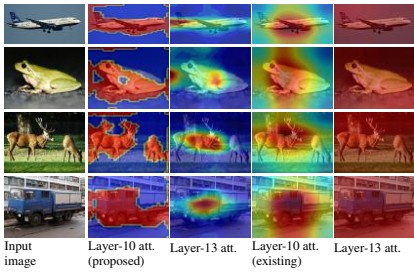

Input image | Layer-10 att. (proposed) | Layer-13 att. | Layer-10 att. (existing) | Layer-13 att.

Figure 3: Attention maps from VGG-att2 trained on low-res CIFAR-10 dataset focus sharply on the objects in high-res ImageNet images of CIFAR categories; contrasted here with the activation-based attention maps of Zagoruyko & Komodakis (2016).

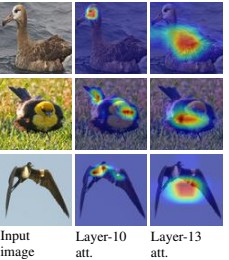

Input image | Layer-10 att. | Layer-13 att.

Figure 4: VGG-att2 trained on CUB-200 for fine-grained bird recognition task: layer-10 learns to fixate on eye and beak regions, layer-13 on plumage and feet.

Within the standard VGG architecture, the proposed attention mechanism provides noticeable performance improvement over baseline models (*e.g.* VGG, RN) and existing attention mechanisms (*e.g.* GAP, PAN) for visual recognition tasks, as seen in Table 1 & 2. Specifically, the *VGG-att2-concat-pc* model achieves a 2.5% and 7.4% improvement over baseline VGG for CIFAR-10 and CIFAR-100 classification, and 7.8% and 0.5% improvement for fine-grained recognition of CUB and SVHN categories. As is evident from Fig. 3, the attention mechanism enables the network to focus on the object of interest while suppressing the background regions. For the task of fine-grained recognition, different layers learn specialised focus on different object parts as seen in Fig. 4. Note

that the RN-34 model for CUB from Table 2 is pre-trained on ImageNet. In comparison, our networks are pre-trained using the much smaller and less diverse CIFAR-100. In spite of the low training cost, our networks are on par with the former in terms of accuracy. Importantly, despite the increase in the total network parameters due to the attention units, the proposed networks generalise exceedingly well to the test set. We are unable to compare directly with the CUB result of Jaderberg et al. (2015) due to a difference in dataset pre-processing. However, we improve over PAN (Seo et al., 2016) by $4.5\%$, which has itself been shown to outperform the former at a similar task. For the remaining experiments, *concat-pc* is our implicit attention design unless specified otherwise. When the same attention mechanism is introduced into RNs we observe a marginal drop in performance: $0.9\%$ on CIFAR-10 and $1.5\%$ on CIFAR-100. It is possible that the skip-connections in RNs work in a manner similar to the proposed attention mechanism, *i.e.* by allowing select local features from earlier layers to skip through and influence inference. While this might make the performance improvement due to attention redundant, our method, unlike the skip-connections, is able to provide explicit attention maps that can be used for auxiliary tasks such as weakly supervised segmentation.

Finally, the global feature vector is used as a query in our attention calculations. By changing the query vector, one could expect to affect the predicted attention pattern. A brief investigation of the extent to which the two compatibility functions allow for such post-hoc control is provided in §A.5.

## 5.2 ROBUSTNESS TO ADVERSARIAL ATTACK

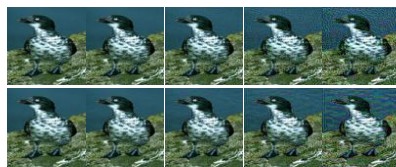

| $L_\infty$ norm | Fooling rate | |
|---|---|---|
| | VGG | VGG-att2 |
| 1 | 58.58 | 53.78 |
| 2 | 78.27 | 76.00 |
| 4 | 89.99 | 89.90 |
| 8 | 94.89 | 95.27 |
| 16 | 96.46 | 97.60 |

Figure 5: Adversarial versions of a sample input image for log-linearly increasing $L_\infty$ norm from 1 to 16, estimated for VGG and VGG-att2 trained on CUB-200.

Figure 6: Network fooling rate measured as a percentage change in the predicted class labels w.r.t those predicted for the unperturbed images.

From Fig. 6, the fooling rate of attention-aware VGG is $5\%$ less than the baseline VGG at an $L_\infty$ noise norm of 1. As the noise norm increases, the fooling rate saturates for the two networks and the performance gap gradually decreases. Interestingly, when the noise begins to be perceptible (see Fig. 5, col. 5), the fooling rate of VGG-att2 is around a percentage higher than that of VGG.

## 5.3 CROSS-DOMAIN IMAGE CLASSIFICATION

| Model | Top-1 accuracies using models trained on CIFAR-10 / CIFAR-100. | | | | | | |
|---|---|---|---|---|---|---|---|
| | STL-train | STL-test | Caltech-101 | Caltech-256 | Event-8 | Action-40 | Scene-67 |
| — Existing archi. — | | | | | | | |
| VGG (Simonyan & Zisserman, 2014) | 45.34 / – | 44.91 / – | 35.97 / 54.20 | 13.16 / 25.57 | 53.62 / 57.05 | 13.85 / 17.58 | 11.02 / 16.73 |
| VGG-GAP (Zhou et al., 2016) | 43.24 / – | 42.76 / – | 41.68 / 62.61 | 16.30 / 31.39 | 58.83 / 68.11 | 16.73 / 24.50 | 12.85 / 23.04 |
| VGG-PAN (Seo et al., 2016) | 47.5 / – | **47.21** / – | 48.09 / 65.75 | 19.61 / 33.66 | 56.93 / 64.49 | 16.96 / 23.44 | 18.77 / 23.43 |
| RN-164 (He et al., 2015) | **47.82** / – | 47.02 / – | 49.89 / 73.62 | 22.59 / 39.65 | 69.19 / 75.10 | 20.56 / 28.72 | 20.26 / 29.89 |
| — Attention archi. — | | | | | | | |
| VGG-att2 | **48.76** / – | 48.29 / – | 55.96 / 74.40 | 26.54 / 41.55 | 67.73 / 80.24 | 22.58 / 29.95 | 25.43 / 30.53 |
| VGG-att3 | 48.42 / – | **48.32** / – | 58.34 / 75.39 | 29.99 / 44.14 | 77.06 / 82.08 | 26.75 / 30.96 | 26.86 / 33.72 |
| RN-164-att2 | 46.36 / – | 46.45 / – | **68.11 / 79.17** | **36.33 / 46.20** | **80.37 / 83.67** | **30.47 / 31.45** | **30.46 / 34.39** |

Table 3: Cross-domain classification: Top-1 accuracies using models trained on CIFAR-10/100.

CIFAR images cover a wider variety of natural object categories compared to those of SVHN and CUB. Hence, we use these to train different network architectures and use the networks as off-the-shelf feature extractors to evaluate their generalisability to new unseen datasets. From Table 3, attention-aware models consistently improve over the baseline models, with an average margin of $6\%$. We make two additional observations. Firstly, low-resolution CIFAR images contain useful visual properties that are transferrable to high-resolution images such as the $600 \times 600$ images of the Event-8 dataset (Li & Fei-Fei, 2007). Secondly, training for diversity is better. CIFAR-10 and CIFAR-100 datasets contain the same corpus of images, only organised differently into 10 and 100 categories respectively. From the results in Table 3, and the attention maps of Fig. 7, it appears that while learning to distinguish a larger set of categories the network is able to highlight more

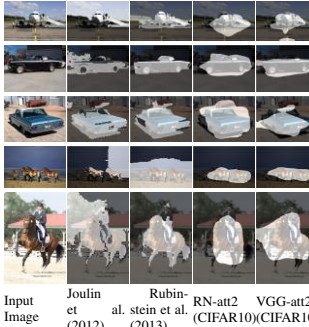

Figure 7: Row 1 : Event-8 (croquet), Row 2 : Scene-67 (bar). Attention maps for models trained on CIFAR-100 (c100) are more diverse than those from the models trained on CIFAR-10 (c10). Note the sharp attention maps in col. 7 versus the uniform ones in col. 4. Attention maps at lower levels appear to attend to part details (*e.g.* the stack of wine bottles in the bar (row 2)) and at a higher level on whole objects owing to a large effective receptive field.

nuanced image regions and capture features that better classify new datasets. (Note that the STL dataset shares the same category set as CIFAR-10. Hence, attention-mounted and baseline models are directly evaluated on STL without the use of an SVM.)

## 5.4 WEAKLY SUPERVISED SEMANTIC SEGMENTATION

| Models | Airplane | Car | Horse |
|---|---|---|---|
| — Attention-based — | | | |
| VGG-GAP (Zhou et al., 2016) | 10.10 / – | 19.81 / – | 29.55 / – |
| VGG-PAN (Seo et al., 2016) | 31.91 / 31.24 | 28.21 / 27.45 | 25.57 / 24.41 |
| VGG-att2 | 34.98 / 31.45 | 28.96 / 38.99 | 29.29 / 29.21 |
| VGG-att3 | **48.07** / 35.66 | 61.19 / 41.23 | **40.95** / 29.68 |
| RN-164-att2 | 41.01 / 45.46 | 63.12 / **65.05** | 36.78 / 40.36 |
| — Saliency-based — | | | |
| Jiang *et al.* (Jiang et al., 2013) | 37.22 | **55.22** | **47.02** |
| Zhang *et al.* (Zhang & Sclaroff, 2013) | **51.84** | 46.61 | 39.52 |
| — Top object proposal-based — | | | |
| MCG (Arbeláez et al., 2014) | **32.02** | **54.21** | **37.85** |
| — Joint segmentation-based — | | | |
| Joulin *et al.* (Joulin et al., 2010) | 15.36 | 37.15 | 30.16 |
| Object-discovery (Rubinstein et al., 2013) | 55.81 | 64.42 | 51.65 |
| Chen *et al.* (Chen et al., 2014) | 54.62 | **69.20** | 44.46 |
| Jain *et al.* (Dutt Jain & Grauman, 2016) | **58.65** | 66.47 | **53.57** |

Figure 8: Weakly supervised segmentation by binarising attention maps.

Figure 9: Jaccard scores (higher is better) for binarised attention maps from CIFAR-10/100 trained models tested on the Object Discovery dataset.

From Table 9, the proposed attention maps perform significantly better at weakly supervised segmentation than those obtained using the existing attention methods (Zhou et al., 2016; Seo et al., 2016) and compare favourably to the top object proposal method, outperforming for all three categories by a minimum margin of 11% and 3% respectively. We do not compare with the CNN-based object proposal methods as they are trained using additional bounding box annotations. We surpass the saliency-based methods in the car category, but perform less well for the other two categories of airplane and horse. This could be due to the detailed structure and smaller size of objects of the latter two categories, see Fig. 8. Finally, we perform single-image inference and yet compare well to the joint inference methods using a group of test images for segmenting the common object category.

## 6 CONCLUSION

We propose a trainable attention module for generating probabilistic landscapes that highlight where and in what proportion a network attends to different regions of the input image for the task of classification. We demonstrate that the method, when deployed at multiple levels within a network, affords significant performance gains in classification of seen and unseen categories by focusing on the object of interest. We also show that the attention landscapes can facilitate weakly supervised segmentation of the predominant object. Further, the proposed attention scheme is amenable to popular post-processing techniques such as conditional random fields for refining the segmentation masks, and has shown promise in learning robustness to certain kinds of adversarial attacks.

**Acknowledgements.** This work was supported by the EPSRC, ERC grant ERC-2012-AdG 321162-HELIOS, EPSRC grant Seebibyte EP/M013774/1 and EPSRC/MURI grant EP/N019474/1.

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

# A    APPENDICES

## A.1    DATASETS

We evaluate the proposed attention models on CIFAR-10 (Krizhevsky & Hinton, 2009), CIFAR-100 (Krizhevsky & Hinton, 2009), SVHN (Netzer et al., 2011) and CUB-200-2011 (Wah et al., 2011) for the task of classification. We use the attention-incorporating VGG model trained on CUB-200-2011 for investigating robustness to adversarial attacks. For cross-domain classification, we test on 6 standard benchmarks including STL (Coates et al., 2010), Caltech-256 (Griffin et al., 2007) and Action-40 (Yao et al., 2011). We use the Object Discovery dataset (Rubinstein et al., 2013) for evaluating weakly supervised segmentation performance. A detailed summary of these datasets can be found in Table 4.

For all datasets except SVHN and CUB, we perform mean and standard deviation normalisation as well as colour normalisation. For CUB-200-2011, the images are cropped using ground-truth bounding box annotations and resized. For cross-domain image classification we downsample the input images to avoid the memory overhead.

| Dataset | Size (total/train/test/extra) | Number of classes / Type | Resolution | Tasks |
|---|---|---|---|---|
| CIFAR-10 (Krizhevsky & Hinton, 2009) | 60,000 / 50,000 / 10,000 / – | 10 / natural images | 32x32 | C, C-c, S |
| CIFAR-100 (Krizhevsky & Hinton, 2009) | 60,000 / 50,000 / 10,000 / – | 100 / natural images | 32x32 | C, C-c, S |
| SVHN (Netzer et al., 2011) | – / 73,257 / 26,032 / 531,131 | 10 / digits | 32x32 | C |
| CUB-200-2011 (Wah et al., 2011) | – / 5,994 / 5,794 / – | 200 / bird images | 80x80 | C |
| STL (Coates et al., 2010) | – / 5,000 / 8,000 / – | 10 / ImageNet images | 96x96 | C-c |
| Caltech-101 (Fei-Fei et al., 2006) | 8677 / – / – / – | 101 / natural images | ∼300x200 | C-c |
| Caltech-256 (Griffin et al., 2007) | 29,780/ – / – / – | 256 / natural images | ∼300x200 | C-c |
| Event-8 (Li & Fei-Fei, 2007) | 1574 / – / – / – | 8 / in/out door sports | >600x600 | C-c |
| Action-40 (Yao et al., 2011) | 9532/ – / – / – | 40 / natural images | ∼400x300 | C-c |
| Scene-67 (Quattoni & Torralba, 2009) | 15613 / – / – / – | 67 / indoor scenes | ∼500x300 | C-c |
| Object Discovery (Rubinstein et al., 2013) | 300/ – / 300 / – | 3 / synthetic and natural images | ∼340x240 | C-c, S |

Table 4: Summary of datasets used for experiments across different tasks (C: classification, C-c: classification cross-domain, S: segmentation).The natural images span from those of objects in plain background to cluttered indoor and outdoor scenes. The objects vary from simple digits to humans involved in complex activities.

## A.2    NETWORK ARCHITECTURES

*Progressive attention networks:* We experiment with the progressive attention mechanism proposed by Seo et al. (2016) as part of our 2-level attention-based VGG models. The attention at the lower level (layer-10) is implemented using a 2D map of compatibility scores, obtained using the parameterised compatibility function discussed in §3.2. Note that at this level, the compatibility scores are not jointly normalised using the softmax operation, but are normalised independently using the pointwise sigmoid function. These scores, at each spatial location, are used to weigh the corresponding local features before the feature vectors are fed to the next network layer. This is the filtering operation at the core of the progressive attention scheme proposed by Seo et al. (2016). For the final level, attention is implemented in the same way as in *VGG-att2-concat-pc*. The compatibility scores are normalised using a softmax operation and the local features, added in proportion to the normalised attention scores, are trained for image classification.

We implement and evaluate the above-discussed progressive attention approach as well as the proposed attention mechanism with the VGG architecture using the codebase provided here: https://github.com/szagoruyko/cifar.torch. The code for CIFAR dataset normalisation is included in the repository.

*Attention in ResNet:* For the ResNet architecture, we make the attention-related modifications to the network specification provided here: https://github.com/szagoruyko/wide-residual-networks/tree/fp16/models. The baseline ResNet implementation consists of 4 distinct levels that project the RGB input onto a 256-dimensional space through 16-, 64-, and 128-dimensional embedding spaces respectively. Each level excepting the first, which contains 2 convolutional layers separated by a non-linearity, contains $n$-residual blocks. Each residual block in turn contains a maximum of 3 convolutional layers interleaved with non-linearities. This yields a network definition of

$9n + 2$ parameterised layers (He et al., 2015). We work with an $n$ of 18 for a 164-layered network. Batch normalization is incorporated in a manner similar to other contemporary networks.

We replace the spatial average pooling layer after the final and $4^{th}$ level by convolutional and max-pooling operations which gives us our global feature vector $g$. We refer to the network implementing attention at the output of the last level as RN-att and with attention at the output of last two levels as RN-att2. Following the results obtained on the VGG network, we train the attention units in the *concat-pc* framework.

## A.3 Training routines

VGG networks for CIFAR-10, CIFAR-100 and SVHN are trained from scratch. We use a stochastic gradient descent (SGD) optimiser with a batch size of 128, learning rate decay of $10^{-7}$, weight decay of $5 \times 10^{-4}$, and momentum of 0.9. The initial learning rate for CIFAR experiments is 1 and for SVHN is 0.1. The learning rate is scaled by 0.5 every 25 epochs and we train over 300 epochs for convergence. For CUB, since the training data is limited, we initialise the model with the weights learned for CIFAR-100. We use the transfer-learning training schedule inspired by Redmon et al. (2015). Thus the training starts at a learning rate of 0.1 for first 30 epochs, is multiplied by 2 twice over the next 60 epochs, and then scaled by 0.5 every 30 epochs for the next 200 epochs.

For ResNet, the networks are trained using an SGD optimizer with a batch size of 64, initial learning rate of 0.1, weight decay of $5 \times 10^{-4}$, and a momentum of 0.9. The learning rate is multiplied by 0.2 after 60, 120 and 160 epochs. The network is trained for 200 epochs until convergence. We train the models for CIFAR-10 and CIFAR-100 from scratch.

All models are implemented in Torch and trained with an NVIDIA Titan-X GPU. Training takes around one to two days depending on the model and datasets.

## A.4 Task-specific processing

We generate the adversarial images using the fast gradient sign method of Goodfellow et al. (2014) and observe the network fooling behaviour at increasing $L_\infty$ norms of the perturbations.

For cross-domain classification, we extract features at the input of the final fully connected layer of each model, use these to train a linear SVM with $C = 1$ and report the results of a 5-fold cross validation, similar to the setup used by Zhou et al. (2016). At no point do we fine-tune the networks on the target datasets. We perform ZCA whitening on all the evaluation datasets using the pre-processing Python scripts specified in the following for whitening CIFAR datasets: https://github.com/szagoruyko/wide-residual-networks/tree/fp16.

For weakly supervised segmentation, the evaluation datasets are preprocessed for colour normalisation using the same scheme as adopted for normalising the training datasets of the respective models. For the proposed attention mechanism, we combine the attention maps from the last 2 levels using element-wise multiplication, take the square root of the result to re-interpret it as a probability distribution (in the limit of the two probabilistic attention distributions approaching each other), rescale the values in the resultant map to a range of $[0, 1]$ and binarise the map using the Otsu binarization threshold. For the progressive attention mechanism of Seo et al. (2016), we simply multiply the attention maps from the two different levels without taking their square root, given that these attention maps are not complementary but sequential maps used to fully develop the final attention distribution. The rest of the operations of magnitude rescaling and binarisation are commonly applied to all the final attention maps, including those obtained using GAP (Zhou et al., 2016).

## A.5 Query-driven attention patterns

In our framework, the global feature vector is used as a query for estimating attention on the local image regions. Thus, by changing the global feature vector, one could expect to affect the attention distribution estimated over the local regions in a predictable manner. The extent to which the two different compatibility functions, the parameterised and the dot-product-based compatibility functions, allow for such external control over the estimated attention patterns may be varied.

For the purpose of the analysis, we consider two different network architectures. The first is the *(VGG-att2)-concat-dp* (DP) model from Table 1 which uses a dot-product-based compatibility function. The other is the *(VGG-att3)-concat-pc* (PC) model which makes use of the parameterised compatibility function. In terms of the dataset, we make use of the extra cosegmentation image set available with the Object Discovery dataset package. We select a single image centrally focused on an instance of a given object category and call this the query image. We then gather a few distinct images that contain objects of the same category but in a more cluttered environment with pose and intra-class variations. We call these the target images.

In order to visualise the role of the global feature vector in driving the estimated attention patterns, we perform two rounds of experiments. In the first round, we obtain both the global and local image feature vectors from a given target image, shown in column 2 of every row of Figure 10. The processing follows the standard protocol and the resulting attention patterns at layer 10 for the two architectures can be seen in columns 3 and 6 of the same figure. In the second round, we obtain the local feature vectors from the target image but the global feature vector is obtained by processing the query image specific to the category being considered, shown in column 1. The new attention patterns are displayed in columns 4 and 7 respectively. The changes in the attention values at different spatial locations as a proportion of the original attention pattern values are shown in columns 5 and 8 respectively.

Notably, for the dot-product-based attention mechanism, the global vector plays a prominent role in guiding attention. This is visible in the increase in the attention magnitudes at the spatial locations near or related to the query image object. On the other hand, for the attention mechanism that makes use of the parameterised compatibility function, the global feature vector seems to be redundant. Any change in the global feature vector does not transfer to the resulting attention map. In fact, numerical observations show that the magnitudes of the global features are often a couple of orders of magnitude smaller than those of the corresponding local features. Thus, a change in the global feature vector has little to no impact on the predicted attention scores. Yet, the attention maps themselves are able to consistently highlight object-relevant image regions. Thus, it appears that in the case of parameterised compatibility based attention, the object-centric high-order features are learned as part of the weight vector $u$. These features are adapted to the training dataset and are able to generalise to new images inasmuch as the object categories at test time are similar to those seen during training.

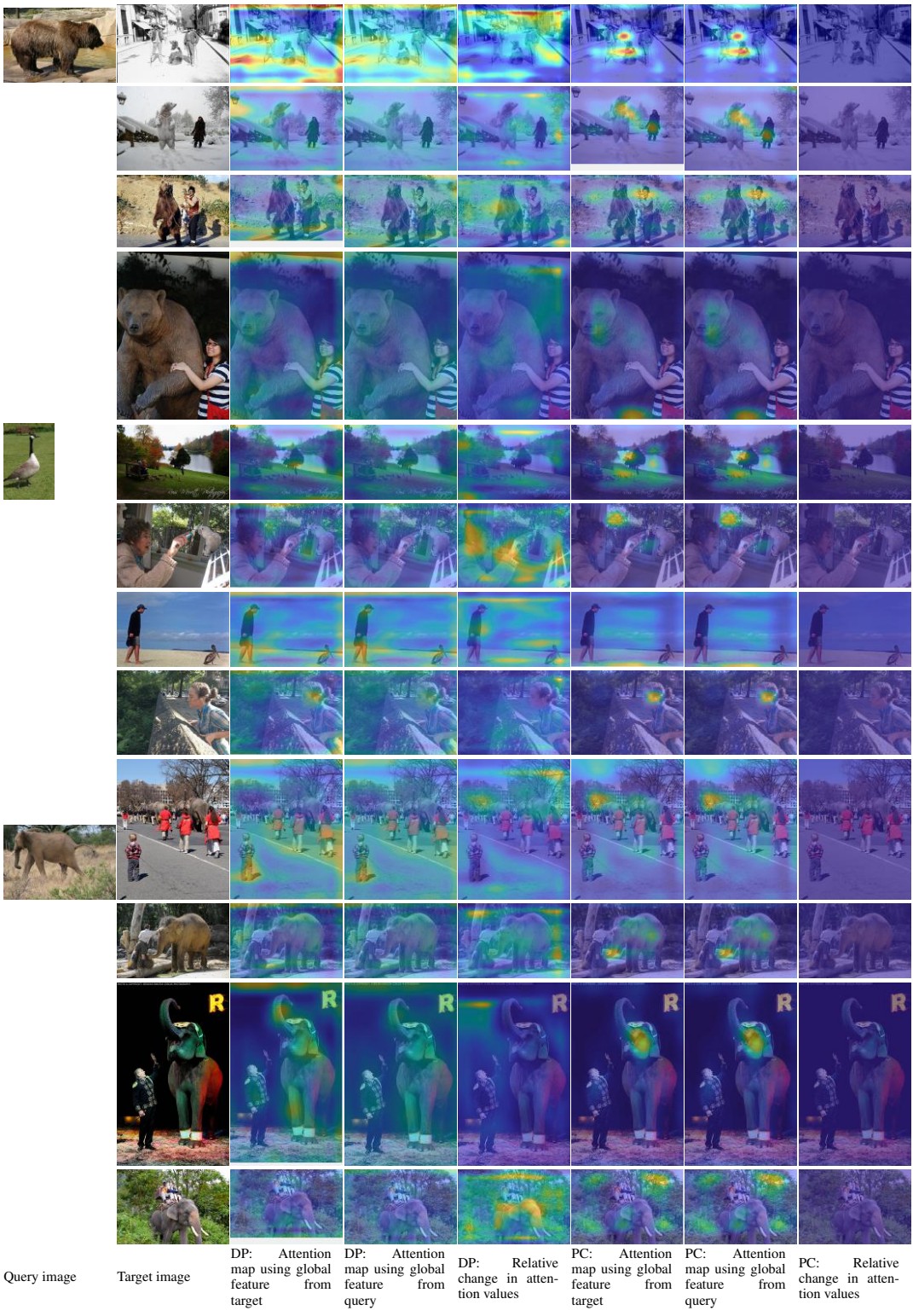

Figure 10: Visual analysis of how a global feature vector obtained from a query image affects the attention patterns on the local image regions of another distinct target image.

