# OpenReview forum: "Learn to Pay Attention"
_ICLR.cc/2018/Conference — Accept (Poster)_

### Official Review · AnonReviewer1 · 2017-11-27
**Has interesting idea but needs improvement**

**Rating:** 5
**Confidence:** 4

**Review:**

This paper proposes a network with the standard soft-attention mechanism for classification tasks, where the global feature is used to attend on multiple feature maps of local features at different intermediate layers of CNN. The attended features at different feature maps are then used to predict the final classes by either concatenating features or ensembling results from individual attended features. The paper shows that the proposed model outperforms the baseline models in classification and weakly supervised segmentation.

Strength:
- It is interesting idea to use the global feature as a query in the attention mechanism while classification tasks do not naturally involve a query unlike other tasks such as visual question answering and image captioning.

- The proposed model shows superior performances over GAP in multiple tasks.

Weakness:
- There are a lot of missing references. There have been a bunch of works using the soft-attention mechanism in many different applications including visual question answering [A-C], attribute prediction [D], image captioning [E,F] and image segmentation [G]. Only two previous works using the soft-attention (Bahdanau et al., 2014; Xu et al., 2015) are mentioned in Introduction but they are not discussed while other types of attention models (Mnih et al., 2014; Jaderberg et al., 2015) are discussed more.

- Section 2 lacks discussions about related work but is more dedicated to emphasizing the contribution of the paper.

- The global feature is used as the query vector for the attention calculation. Thus, if the global feature contains information for a wrong class, the attention quality should be poor too. Justification on this issue can improve the paper.

- [H] reports the performance on the fine-grained bird classification using different type of attention mechanism. Comparison and justification with this method can improve the paper. The performance in [H] is almost 10 % point higher accuracy than the proposed model.

- In the segmentation experiments, the models are trained on extremely small images, which is unnatural in segmentation scenarios. Experiments on realistic settings should be included. Moreover, [G] introduces a method of using an attention model for segmentation, while the paper does not contain any discussion about it.


Overall, I am concerned that the proposed model is not well discussed with important previous works. I believe that the comparisons and discussions with these works can greatly improve the paper.

I also have some questions about the experiments:
- Is there any reasoning why we have to simplify the concatenation into an addition in Section 3.2? They are not equivalent.

- When generating the fooling images of VGG-att, is the attention module involved, or do you use the same fooling images for both VGG and VGG-att?

Minor comments:
- Fig. 1 -> Fig. 2 in Section 3.1. If not, Fig. 2 is never referred.

References
[A] Huijuan Xu and Kate Saenko. Ask, attend and answer: Exploring question-guided spatial attention for visual question answering. In ECCV, 2016.
[B] Zichao Yang, Xiaodong He, Jianfeng Gao, Li Deng, and Alex Smola. Stacked attention networks for image question answering. In CVPR, 2016.
[C] Jacob Andreas, Marcus Rohrbach, Trevor Darrell, and Dan Klein. Deep compositional question answering with neural module networks. In CVPR, 2016.
[D] Paul Hongsuck Seo, Zhe Lin, Scott Cohen, Xiaohui Shen, and Bohyung Han. Hierarchical attention networks. arXiv preprint arXiv:1606.02393, 2016.
[E] Quanzeng You, Hailin Jin, Zhaowen Wang, Chen Fang, and Jiebo Luo. Image captioning with semantic attention. In CVPR, 2016.
[F] Jonghwan Mun, Minsu Cho, and Bohyung Han. Text-Guided Attention Model for Image Captioning. AAAI, 2017.
[G] Seunghoon Hong, Junhyuk Oh, Honglak Lee and Bohyung Han, Learning Transferrable Knowledge for Semantic Segmentation with Deep Convolutional Neural Network, In CVPR, 2016.
[H] Max Jaderberg, Karen Simonyan, Andrew Zisserman, Koray Kavukcuoglu, Spatial Transformer Networks, NIPS, 2015

---

> ### Author Response · Authors · 2017-12-19
> **Response to Reviewer: Has interesting idea but needs improvement**
>
> We thank the reviewer for the comments.
>
> Please find as follows, our point-by-point response to the concerns raised in the weakness section.
>
> 1 and 2 - missing references and discussion about related work: We thank the reviewer for pointing us to the most recent relevant literature regarding the proposed attention scheme. We have provided a brief discussion of the suggested works in the third paragraph of Sec. 1. A more thorough treatment is taken up in Sec. 2 (Related Work), which has now been reorganised to more exhaustively capture the variety of existing approaches in the area of attention in deep neural networks.
> We have also produced an experimental comparison against the progressive attention mechanism of Seo et al., incorporated into the VGG model and trained using the global feature as the query, for the task of classification of CIFAR datasets. The details of the implementation are provided in appendix Sec. A.2. The results are compiled in the updated Table 1. A quantitative evaluation of the above mechanism for the task of fine-grained recognition on the CUB and SVHN datasets is forthcoming and will be made available in the next revision.
>
> 3 - the global feature as a driver of attention: The global feature is indeed used as the query vector for our attention calculations. The global and local features vector don't always need to be obtained from the same input image. In fact, in our framework, we can extract the local features from a given image A: call that the target image. The global feature vector can be obtained from another image B, which we call the query image. Under the proposed attention scheme, it is expected that the attention maps will highlight objects in the target image that are 'similar' to the query image object. The precise notion of 'similarity' may be different for the two compatibility functions, where parameterised compatibility is likely to capture the concept of objectness while the dot-product compatibility is likely to learn a high-order appearance-based match between the query and target image objects.  We are investigating the two different compatibility functions w.r.t. the above hypothesis. The experimental results in the form of visualisations will be made available in the next update.
>
> 4 - performance comparison with [H]: We are unable to compare with [H] for the task of fine-grained recognition due to a difference in dataset preprocessing. The CUB dataset in [H] has not been tighly cropped to the birds in the images. Thus, the network has access to the background information which in case of birds can offer useful information about their habitat and climate, something that is key to their classification.
> However, note that our experimental comparison with the progressive attention mechanism of Seo et al. is forthcoming. The progressive attention scheme has been shown to outperform [H] for the task of attribute prediction, see Table 1 in [D]. Hence, by presenting a comparison with the former, we would be able to indirectly compare the proposed approach against the spatial transformer network architecture of [H].
>
> 5 - segmentation experiments and comparison: Our weakly supervised segmentation experiments make use of the Object Discovery dataset, a known benchmark in the community widely used for evaluating approaches developed for the said task [I, J]. We note that [G] presents an attention-based model for segmentation. Our work uses category labels for weak segmentation and is related to the soft-attention approach of [G]. However, unlike the aformentioned, we do not explicitly train our model for the task of segmentation using any kind of pixel-level annotations. We evaluate the binarised spatial attention maps, learned as a by-product of training for image classification, for their ability to segment objects. We have added this discussion to the paragraph on weakly supervised segmentation in Sec. 2 (Related Work).
>
>
> Our responses to the questions asked are below:
>
> 1. concatenation vs addition: Given the existing free parameters between the local and the global image descriptors in a CNN pipeline, we can simplify the concatenation of the two descriptors to an addition operation, without loss of generality. This allows us to limit the parameters of the attention unit.
>
> 2. generation of fooling images: When generating the fooling images of VGG-att, we do use the attention module. Thus, the fooling images for both VGG and VGG-att are conditioned on their respective architectures, and hence different.
>
>
> We have incorporated the minor comment in the updated version.
>
>
> References
> [I] Dutt Jain, S., & Grauman, K. (2016). Active image segmentation propagation. In Proceedings of the IEEE Conference on Computer Vision and Pattern Recognition (pp. 2864-2873).
> [J] Rubinstein, M., Liu, C., & Freeman, W. T. (2016). Joint Inference in Weakly-Annotated Image Datasets via Dense Correspondence. International Journal of Computer Vision, 119(1), 23-45.

---

### Official Review · AnonReviewer2 · 2017-11-27
**No title**

**Rating:** 6
**Confidence:** 4

**Review:**

This paper proposed an end-to-end trainable hierarchical attention mechanism for CNN. The proposed method computes 2d spatial attention map at multiple layers in CNN, where each attention map is obtained by computing compatibility scores between the intermediate features and the global feature. The proposed method demonstrated noticeable performance improvement on various discriminative tasks over existing approaches.

Overall, the idea presented in the paper is simple yet solid, and showed good empirical performance. The followings are several concerns and suggestions.

1. The authors claimed that this is the first end-to-end trainable hierarchical attention model, but there is a previous work that also addressed the similar task:
Seo et al, Progressive Attention Networks for Visual Attribute Prediction, in Arxiv preprint:1606.02393, 2016

2. The proposed attention mechanism seems to be fairly domain (or task ) specific, and may not be beneficial for strong generalization (generalization over unseen category). Since this could be a potential disadvantage, some discussions or empirical study on cross-category generalization seems to be interesting.

3. The proposed attention mechanism is mainly demonstrated for single-class classification task, but it would be interesting to see if it can also help the multi-class classification (e.g. image classification on MS-COCO or PASCAL VOC datasets)

4. The localization performance of the proposed attention mechanism is evaluated by weakly-supervised semantic segmentation tasks. In that perspective, it would be interesting to see the comparisons against other attention mechanisms (e.g. Zhou et al 2016) in terms of localization performance.

---

> ### Author Response · Authors · 2017-12-19
> **Response to Reviewer**
>
> We thank the reviewer for the comments.
> Our responses are provided below, numbered correspondingly:
>
> 1. The missing comparison with the progressive attention mechanism of Seo et al. was unintentional, and the authors agree that their work is indeed closely related to the proposed work. We have now included a discussion on it in Sec. 2 (Related Work). We have also produced an experimental comparison against the progressive attention mechanism, incorporated into the VGG architecture and trained using the global feature as the query, for the task of classification of CIFAR datasets. The details of the implementation are provided in appendix Sec. A.2. The results are compiled in the updated Table 1. A quantitative evaluation of the above mechanism for the task of fine-grained recognition on the CUB and SVHN datasets is forthcoming and will be made available in the next revision.
>
> 2. For our detailed investigation of cross-category generalisation of the image features learned using the proposed attention scheme, we would like to point the reviewer to Sec. 5.3. Here, we use the baseline and attention-enhanced models as off-the-shelf feature extractors. The models are trained for the tasks of classification on CIFAR datasets and are queried to obtain high-order representations of images from unseen datasets, such as the Action-40 and Scene-67 datasets. At train time, the training-set features are used to optimise a linear SVM as a classifier. At test time, we evaluate the quality of generalisation via the quality of classification of the test set based on the extracted features.
>
> 3. The multi-class classification task can be posed as a set of single-class or one-hot-encoded classification tasks. In this regard, as confirmed experimentally, our proposed attention scheme would be able to offer performance benefits. However, a more direct multi-class classification, on datasets such as MS-COCO or PASCAL, can be covered in future versions given that the standard protocol to train and test such networks is typically lengthy. The models are usually pre-trained on ImageNet and fine-tuned on the above datasets, putting this a bit outside our scope for revision/addition at present.
>
> 4. The comparison with the attention mechanism proposed by Zhou et al. was actually included in Fig. 9 ("VGG-GAP"): the reference was missing, which we have now added. The model has been trained for the CIFAR-10 dataset, as the classes of this dataset overlap with the Object Discovery dataset that includes car, horse and airplane as categories.

---

### Official Review · AnonReviewer3 · 2017-11-29
**Results are good, some unclear explanation**

**Rating:** 6
**Confidence:** 4

**Review:**

This paper proposes an end-to-end trainable attention module, which takes as input the 2D feature vector map and outputs a 2D matrix of scores for each map. The goal is to make the learned attention maps highlight the regions of interest while suppressing background clutter. Experiments conducted on image classification and weakly supervised segmentation show the effectiveness of the proposed method.

Strength of this paper:
1) Most previous work are all implemented as post-hoc additions to fully trained networks while this work is end-to-end trainable. Not only the newly added weights for attention will be learned, so are the original weights in the network.
2) The generalization ability shown in Table 3 is very good, outperforming other existing network by a large margin.
3) Visualizations shown in the paper are convincing.

Some weakness:
1) Some of the notations are unclear in this paper, vector should be bold, hard to differentiate vector and scalar.
2) In equation (2), l_i and g should have different dimensionality, how does addition work? Same as equation (3)
3) The choice of layers to add attention modules is unclear to me. The authors just pick three layers from VGG to add attention, why picking those 3 layers? Is it better to add attention to lower layers or higher layers? Why is it the case that having more layers with attention achieves worse performance?

---

> ### Public Comment · (anonymous) · 2017-12-02
> **Results are good, some unclear explanation**
>
> (Not an author, but)
>
> For equation (2), the authors have stated in the comments that the actual choice for which mapping method to use to have the dimensionalities of local features and the global descriptor align is an implementation detail. They've proposed two methods:
>
> 1. Map the local features whose dimensionalities don't align with that of g to the correct dimensionality through densely connected layers.
>
> 2. Map g to the dimensionalities of local features  through densely connected layers.

---

> ### Author Response · Authors · 2017-12-19
> **Response to Reviewer: Results are good, some unclear explanation**
>
> We thank the reviewer for the comments.
>
> 1) notation: We have updated the paper, in particular Section 3, to represent the vectors in bold to differentiate them from scalars.
>
> 2) potential for differing dimensionalities of l_i and g: There is some discussion of this in the second paragraph of Sec. 3.1. We propose the use of one fully connected layer for each CNN layer s, which projects the local feature vectors of s to the dimensionality of g. These linear parameters are learned along with all other network parameters during end-to-end training. There is an implementation detail, though, which we had neglected to mention: in order to limit the network parameters at the classification stage, we actually project g to the lower-dimensional space of the local features l_i. A note of clarification on this has been added to the first paragraph of Sec. 4.
>
> 3) selection of layers for attention: A brief discussion on the choice of adding attention to higher layers as opposed to the lower ones was included in Sec. 3.3. We have now augmented this discussion, in place, with further clarification on the specific layers that we choose for estimating the attention.
> For l_i and g to be comparable using the proposed compatibility functions, they should be mapped to a common high-dimensional space. In other words, the effective filters operating over image patches in the layers s must represent relatively ‘mature’ features that are captured in g for the classification goal. We thus expect to see the greatest benefit in deploying attention relatively late in the pipeline to provide for the learning of these features in l_i. In fact, att2 architectures often outperform their att3 counterparts, as can be seen in Tables 1 and 2.
> Further, different kinds of class details are more easily accessible at different scales. Thus, in order to facilitate the learning of diverse and complementary attention-weighted features, we propose the use of attention over different spatial resolutions. The combination of the two factors stated above results in our deploying the attention units after the convolutional blocks that are late in the pipeline, but before their corresponding max-pooling operations, i.e. before a reduction in the spatial resolution.

---

### Public Comment · (anonymous) · 2017-11-02
**ResNet Model Architectures**

It's not totally clear what the ResNet model architecture is.  Section A.2 provides some details, but specifically I don't see how you're getting the dimensions of g and the intermediate layers to match up.  Are you able to provide some more detailed information about the architectures?

---

> ### Author Response · Authors · 2017-11-06
> **Learning to match the dimensions of local features and g**
>
> Thank you for your question. There is some discussion of this in paragraph 2, Sec. 3.1 though we should perhaps provide more details. We propose the use of one fully connected layer for each CNN layer s, that projects the local feature vectors of s to the dimensionality of g. These linear parameters are learned along with all other network parameters over the end-to-end training. There is an implementation detail, though, which we've neglected to mention: in order to limit the network parameters at the classification stage, we actually project g to the lower-dimensional space of the local features. We can update the section to reflect this in more detail.

---

### Public Comment · (anonymous) · 2017-11-14
**Dataset downsampling, weight vector in the parametrised compatibility score calculation, local feature source, more on local features and g.**

Is the weight vector used in parametrised compatibility score calculation ('u') the same for every c calculation, or should a separate u vector be trained for each of the three attention submodules?
What target resolution was used for downsampling images in the cross-domain classification datasets?
Are the local features fed to the attention submodules extracted from the outputs of the pooling layers or from the outputs of the convolutional layer preceding them? (For example, in VGG-att3, is L_1 extracted directly from the output of the third 256-filter convolutional layer or from the 2x2 max pooling layer right after it?)
In the comment below, you proposed that in order to align the dimensionality of global g and the local feature vectors for the compatibility calculation step, local features should first be passed through an additional fully connected layer; should this step be done even if the dimensionality of g and local feature vectors already line up? (For example, as in the vectors from L_3 in VGG-att3)
Additionally, you proposed that instead of the local features being mapped to the dimensionality of g, g can be mapped to the dimensionality of the local features; in this case, if there are two or more submodules with local feature vectors of equal dimensionality, should g be mapped to each one separately, or should g only be projected to any given dimensionality once for shared use in the submodules with local features of that dimensionality?

---

> ### Author Response · Authors · 2017-11-17
> **Clarifications of Implementation Details**
>
> Thank you for your questions, answered sequentially below:
>
> The weight vector u used in the parameterised compatibility score calculation corresponds to a given layer s; a different weight vector u_s is learned for each layer. As written, the expression in (2) assumes the context of a layer, and so the s index is implicit. We can make it explicit for better clarity.
>
> The input image resolution of CIFAR-trained models is 32x32x3. This is the size to which the images of the test datasets are downsampled in the cross-domain classification experiments.
>
> The local features are the ReLU-activated outputs of the convolutional layers before the corresponding max-pooling operation. This is done to keep the resolution of the attention maps as high as possible.
>
> The global vector g is mapped to the dimensionality of the local features l (at a given layer s) by a linear layer if and only if their dimensionalities differ. Following from this, yes, the global vector g, once mapped to a given dimensionality, is then shared by the local features from different layers s as long as they are of that dimensionality.

---

### Public Comment · (anonymous) · 2017-11-19
**Learning rate decay during transfer learning, color normalization algorithm**

When training the models on CUB-200-2011 with weights initialized from those learned on CIFAR-100, did you still use 10^-7 as learning rate decay, or did you set learning rate decay to 0 and strictly stick to the exact learning rates from the transfer learning schedule?

Exactly which color normalization algorithm did you use? Also, in what order were dataset-wide mean and standard deviation normalization and color normalization done?

---

> ### Author Response · Authors · 2017-11-25
> **Learning rate decay during transfer learning, color normalization algorithm**
>
> Thank you for your questions.
> When training the models on CUBS-200-2011 with weights initialised from those learned on CIFAR-100, we continue to use 10^-7 as the learning rate decay. Only the schedule for the learning rates is modified as explained.
>
> We use the ZCA whitening method[1,2] for mean, standard deviation and color normalisation widely adopted for the pre-processing of CIFAR datasets.
>
> 1. Goodfellow, Ian J., et al. "Maxout networks." arXiv preprint arXiv:1302.4389 (2013).
> 2. Zagoruyko, Sergey, and Nikos Komodakis. "Wide residual networks." arXiv preprint arXiv:1605.07146 (2016).

---

### Public Comment · (anonymous) · 2017-11-28
**ResNet implementation details**

Does the input go through batch normalization before it is passed to the first convolutional layer?
What number of filters, kernel size and strides are used in the two convolutional layers in level 1?
Is the second convolutional layer in level 1 followed by ReLU too, or is the only activation used in level 1 the ReLU between the two convolutional layers?
Is there a max-pooling layer at the end of each of the four levels in the implementation, including level 1 and level 4? If so, what pool size and strides are used in each?
Does the implementation use the bottleneck design for residual blocks from the original paper? If not, what design is used for the residual blocks?  (So far, we've assumed the bottleneck design, with Conv(16,1)->BN->ReLu->Conv(16,3)->BN->ReLU->Conv(64,1)->BN->identity addition->ReLU in level 2, same in level 3 but using convolutions with parameters (32,1), (32,3), and (128,1), and same in level 4 but using convolution with parameters (64,1), (64,3), and (256,1))

Is every convolutional layer in the model (except the ones used for dimensionality increases) followed by batch normalization, including convolutional layers in level 1?
What number of filters, kernel size, and strides are used for the final convolutional layer? Is this layer followed by batch normalization and ReLU as well?
What pool size and strides are used for the final pooling layer?
Is the flattened output of the final pooling layer directly mapped to the dimensionalities of the local features through three separate linear fully-conected layers, or is it first passed through a ReLU-activated fully connected layer?

Thank you for the answers.

---

> ### Author Response · Authors · 2018-01-05
> **ResNet implementation details**
>
> Thank you for your question.
>
> We use the ResNet implementation provided at the following link as our baseline ResNet model - https://github.com/szagoruyko/wide-residual-networks/tree/fp16/models. As specified earlier, we work with a 164-layered network. This should help to clarify the details regarding the location and specifications of batch normalisation, max-pooling, non-linearity and convolutional operations. This reference has now also been added to the appendix section A.2.
>
> As discussed before, for the ResNet architecture, we incorporate our attention modules at the outputs of the last two levels, i.e. on local feature vectors of dimensionalities 128 and 256, respectively. We remove the spatial averaging step after the final convolutional layer in the original architecture. Instead, we obtain the global feature vector by processing the batch-normalised and ReLU-activated output of the final level using a convolutional layer with a kernel size of 3x3 and 256 output channels, a ReLU non-linearity, and a fully connected 256x256 layer. The convolutional layer is itself sandwiched between two max-pooling layers, that together downsample the input by a factor of 8 in each of the two spatial dimensions, to yield a single 256 dimensional vector. The global feature vector is used directly for estimating attention at the final level where the local features have a dimensionality of 256. For the lower level, with a dimensionality of 128, the global vector is downsized to a matching dimensionality by a single 256x128 fully connected layer.
>
> To add to the above, we will make our implementation of the proposed method public, post an internal review.

---

### Author Response · Authors · 2018-01-01
**Additional comparisons suggested by the reviewers**

Once again, we thank the reviewers for their comments.
We have now added to the paper a complete experimental comparison against the progressive attention approach proposed by Seo et al. We incorporate the progressive attention mechanism at 2 levels in the baseline VGG architecture and evaluate it on the various tasks considered in the paper. The details of the implementation are provided in the appendix.
The results for image classification and fine-grained recognition can be found in Tables 1 and 2 respectively, those for domain shifted classification in Table 3, and those for weakly supervised segmentation in Figure 9. The proposed attention method consistently outperforms the former across the board.
Furthermore, an indirect comparison against the spatial transformer networks of Jaderberg et al. on the task of fine-grained recognition is now included in the discussion in the results section. As we state there, we are unable to compare with the CUB result of Jaderberg et al. directly, due to a difference in dataset pre-processing. However, we improve over the progressive attention approach of Seo et al. by 4.5%, and note that progressive attention has itself been shown to perform better than spatial transformer networks at the similar task of attribute prediction using attention.

---

### Author Response · Authors · 2018-01-05
**Query-driven attention patterns**

In the proposed framework, the global feature is indeed used as the query vector for the attention calculations. Thus, by changing the global feature vector, one could expect to affect the estimated attention patterns in a predictable manner. We have now included in the appendix section A.5., a brief discussion and a qualitative comparison of the extent to which the two different compatibility functions allow for a post-hoc control of the estimated attention scores by influencing the global image vector.

---

### Decision · Program_Chairs · 2018-01-29
**ICLR 2018 Conference Acceptance Decision**

**Decision:**

Accept (Poster)

**Comment:**

quality: interesting idea to train an end-to-end attention together with CNNs and solid experiments to justify the benefits of using such attentions.
clarity: the presentation has been updated according to review comments and improved a lot
significance: highly relevant topic, good improvements over other methods